# Experiences of People with Cancer from Rural and Remote Areas of Western Australia Using Supported Accommodation in Perth While Undergoing Treatment

**Andrette Chua, Evelyn Nguyen, Li Lin Puah, Justin Soong and Sharon Keesing \***

School of Allied Health, Curtin University, Bentley, Perth, WA 6102, Australia;
andrette.chua@graduate.curtin.edu.au (A.C.); evelyn.nguyen@graduate.curtin.edu.au (E.N.);
lilin.puah@graduate.curtin.edu.au (L.L.P.); justin.soong@student.curtin.edu.au (J.S.)
**\*** Correspondence: s.keesing@curtin.edu.au

**Abstract:** The aim of the study was to explore the lived experiences of people diagnosed with cancer from rural and remote areas of Western Australia, who utilise supported accommodation services whilst undergoing treatment in the capital city (Perth). Methods A qualitative phenomenological approach was used in this study. Ten participants were recruited using purposive sampling, who were aged between 35–65 years, were diagnosed with cancer within the previous three months and used accommodation services within the past 12 months. Semi-structured in-depth interviews were conducted with a duration of approximately 45–60 min via Zoom, FaceTime or phone call. Interview data was transcribed, thematically analysed and coded into relevant themes. Results: Three overarching themes were derived from the interviews–"It's harder to have cancer when you have to relocate for treatment," "The paradoxical experience of staying at the accommodation," and "Feeling grateful for the support offered'. Conclusions: People diagnosed with cancer who have to relocate during treatment require emotional, logistical, and social supports. Cancer accommodation services are essential in enabling individuals to continue engaging in meaningful occupations and maintain their quality of life. Our study highlights the need for cancer accommodation services to consider the complex needs of individuals completing treatment for cancer in locations away from their usual homes.

**Keywords:** cancer; lived experiences; rural; accommodation; Cancer Council Western Australia

## 1. Introduction

Cancer is a major cause of illness and death across all ages and the greatest contributor to the fatal burden of disease in Australia [1]. In 2019, there was an estimated incidence rate of 144,713 new cases of cancer in Australia [2]. In Western Australia (WA), 60,009 people were diagnosed between 2010–2014 and 20,510 deaths from cancer were registered between 2012–2016 [2]. People with cancer often experience disruption to their everyday lives due to a range of financial, physical, and psychosocial limitations [3]. These concerns are often exacerbated for those living in rural and remote areas who are required to relocate in order to receive treatment [4]. Approximately 30% of the Australian population live in rural and remote areas, increasing the risk of experiencing poorer health outcomes and lower quality of life, especially with a diagnosis of cancer [5,6]. Despite the federal government establishing and funding $694 million to the Regional Cancer Centre (RCC) initiative, significant inequalities remain regarding access to specialist oncology services in rural and remote areas in Australia, particularly areas that are not in-receipt of RCC funding [7]. Multiple factors contribute to these inequalities, including delayed diagnosis, geographical isolation, inadequate transport [8], which may result in people with cancer needing to relocate in order to access cancer treatment.

Many people diagnosed with cancer who live in rural and remote locations incur additional expenses related to cancer treatment resulting in increased financial demands [6,9]. These include the cost of travel, accommodation, meals and public transport when required to relocate to the city for treatment. There is also a strong positive correlation between living in remote locations and levels of burden [10], as many individuals who live rurally/remotely experience not only cancer related issues such as physical, psychological and emotional difficulties but additional practical stresses due to concerns relating to being absent from their own homes, local community and their usual supports [6,11].

In WA there are very limited accommodation options for individuals with cancer who are required to relocate to Perth for treatment. These options are limited to a small group of not for profit organisations including Cancer Council Western Australia (CCWA) or short-term rental accommodation. Individuals with cancer who live more than 100 kms away from relevant services and are required to relocate to Perth for treatment are eligible to stay at one of the CCWA lodges. The lodges offer subsidised accommodation for both individual and family members. Individuals who are eligible for the Patient Assisted Travel Scheme (PATS) may receive further financial support to assist with accommodation costs and these are then offered free of charge in some instances. There is very limited research about the experiences of people with cancer from rural areas who are required to access treatment in Perth, WA. The aim of this study is to explore the lived experiences of people diagnosed with cancer from rural and remote areas of WA who utilise CCWA accommodation services whilst undergoing treatment in Perth. The findings from this research will assist to ensure unmet needs are addressed and services are developed to ensure these needs are met.

## 2. Materials and Methods

### 2.1. Study Design

Phenomenology is an interpretive form of qualitative research that aims to study the nature of an experience from the perspective of the individual experiencing the phenomenon, known as the "lived experience" [12,13]. Phenomenology allows researchers to understand the impact of illness on the patient's lives as a primary phenomenon and not as a secondary effect of the disease itself [14]. This approach was appropriate for the study as it explores the subjective experiences of people with cancer. Semi-structured in-depth interviews were chosen as the data collection method as these enabled researchers to explore in-depth how individuals with cancer experience the additional burden of relocating for treatment. [15]. An interview guide was developed based on review of the literature and included demographic information, and a range of questions developed to address the aims of the research. These being the challenges experienced by participants due to their treatment and relocation, their current daily routines and how these contrasted with previous routines as well as any benefits associated with their current living situation. A pilot interview was conducted with an individual with research interview experience who provided suggestions and recommendations to improve the flow, timing and terms used in the interview guide which was modified accordingly prior to use with eligible participants.

### 2.2. Participants Recruitment

Ten participants were recruited through purposive sampling. Purposive sampling increases the probability for observing the phenomenon of interest [16]. Participants were approached by a CCWA staff member who discussed the purpose of the research and invited them to be involved. Flyers explaining the study were displayed in the CCWA accommodation. Potential participants were invited to contact the research team via email or phone call and an appointment was made to complete the interview. Participants were eligible to participate if they: have stayed in CCWA accommodation in Perth within the last 12 months, aged 18 to 65 years, diagnosis of cancer within the previous 3 months, able to provide informed consent, and can communicate in English proficiently.

*2.3. Data Collection*

Originally, data collection was planned as in-person interviews at the accommodation service, however due to the restrictions of COVID-19, interviews transitioned to phone or video calls. Two researchers (E.N. and J.S.) completed the interviews which occurred between April and July 2020. A copy of the participant information statement, consent form, and interview questions were emailed to all participants prior to their interview. Due to the restrictions brought about by COVID-19, written consent was difficult to attain, therefore consent was obtained verbally at the beginning of each interview. Interviews were digitally recorded with the participant's consent and stored according to the confidentiality agreement outlined in the participant information statement. Recruitment ceased when no new concepts were identified during data analysis and saturation of the data was considered to have been reached [17].

The trustworthiness of this study was guided by the four criteria outlined by Lincoln and Guba (1985), which include credibility, dependability, confirmability and transferability [18]., An interview guide was used by the researchers to ensure consistency of questions between each participant. Member checking was conducted to allow the researchers to review statements for accuracy and gain a more holistic understanding of the phenomenon [19]. Peer review of the data was conducted in pairs by the researchers under the primary supervisor's guidance, during the thematic analysis process. A thick description of methods was detailed throughout the study to ensure sufficient information was provided to increase transferability of the findings for future studies [20].

*2.4. Data Analysis*

Thematic analysis described by Braun and Clarke (2006) was used to analyse the data [21]. Two researchers independently transcribed the interviews to familiarize themselves with the data and created initial codes. These researchers met to compare and contrast codes, develop themes and refine themes.

After themes were explored and described, all researchers came to a consensus on the most relevant and appropriate labels to describe the themes identified in the data.

*2.5. Ethical Considerations*

Ethical approval was granted from the Curtin Human Research Ethics Committee (HREC) in December 2019. With changes in the method of data collection due to COVID-19, an amendment was submitted then approved on 18 February 2020 (HREC number: HRE 2020-0030). Information about the study (e.g., purpose, de-identification, confidentiality, withdrawal without penalty) was provided to the participants for them to make an informed decision and provide consent prior to participating in the study. While interviews were not expected to create distress, prior to commencing the interviews, participants were provided with a range of strategies should they become distressed. These strategies included taking a break during the interview or rescheduling and providing information about the Cancer Council helpline and counselling services.

## 3. Results

Ten participants were interviewed (refer to Table 1 for demographic information). Most participants were female, were married and aged 55 years and over. Interviews revealed the experiences of people with cancer who lived in CCWA accommodation while undergoing treatment in Perth, resulting in the three overarching themes and eight sub-themes identified below. The themes presented are categorized chronologically and report participants' views regarding their experiences at the start of treatment, during their time spent at the accommodation, and then following treatment.

**Table 1.** Participant Demographics.

| | P1 | P2 | P3 | P4 | P5 | P6 | P7 | P8 | P9 | P10 |
|---|---|---|---|---|---|---|---|---|---|---|
| Age range (years) | 55 and over | 55 and over | 35–44 | 55 and over | 55 and over | 55 and over | 55 and over | 45–54 | 55 and over | 55 and over |
| Gender | Female | Female | Female | Female | Female | Male | Female | Female | Female | Male |
| Distance from capital city (Perth) | 750 km | 1550 km | 2000 km | 600 km | 200 km | 400 km | 2200 km | 700 km | 400 km | 1200 km |
| Relationship status | Married | Married | Married | Married | Married | Married | Divorced | Divorced | Married | Married |
| Living situation | Lives with spouse | Lives with spouse | Lives with spouse and children | Lives with spouse and other family members | Lives with spouse | Lives with spouse and children | Lives alone with pets | Lives with children | Lives with spouse | Lives with spouse |
| Employment status | Not working | Works full time | Works part time | Not working | Not working | Works shift work | Retired | Works full time | Not working | Not working |

*3.1. Theme 1: It's Harder to Have Cancer when You Have to Relocate for Treatment*

Whilst all participants experienced adverse side effects due to their cancer treatment, most accepted these as part of their cancer experience and were determined to make the most of their time away from home. However, many external factors, such as financial and family concerns, exacerbated the cancer experience for most participants, due to needing to relocate for treatment. Participant three describes this as the hardest part of that period.

*"I could say being far away from my family during the main part of my treatment in particular was the hardest part of it...knowing that I couldn't look after my family."*—P3

Conversely, some participants reported a more positive outlook about how these factors affected them, participant six viewed this stage positively- being required to manage a young family and cope with the demands of cancer treatment were somewhat reduced as they were away from their usual residence and childcare responsibilities during that time.

*"It was awesome...My kids were driving me nuts. So, it was great to get away from home. Even though it wasn't really a holiday, it was still a nice break."*—P6

### 3.1.1. Side Effects and Symptoms from Treatment

Participants described a range of treatment-related side effects including fatigue, nausea, appetite loss, brittle nails, peripheral neuropathy and diarrhea. This created a drastic change in energy levels before and after treatment and prevented participants from engaging in their usual routines and activities. Conversely, participants noted that exercise, naturopathy, Chinese medicine and having adequate rest assisted them to alleviate their symptoms.

*"I was vomiting, I couldn't eat, I couldn't drink. Everything just tastes really bad. I lost a lot of weight very quickly and I just went downhill from there. I just got so ill and I wanted to die . . . I just didn't want to live and go through that."*—P9

### 3.1.2. The Impact of Being Away from Home during Treatment

Being away from family and home resulted in feelings of guilt and sadness, particularly if participants had to miss out on significant family events. Loneliness also impacted on energy levels and mood. Many participants described this period as being isolating as they had few friends or family located in the city. Participants two and four describe their feelings of loneliness and missing their usual home life and routines.

*"Naturally, it means leaving, sometimes your family behind. So, it gets a little bit lonely. It gets a bit isolating. We're not from Perth. So, we don't really have many friends in Perth. I know one or two people, but, you know, not many. So, yes, it affects your home life. It affects your job."*—P2

*"So, we didn't know anybody in Perth at that stage. Both of our sons were living out in XXXX and we had no family or friends the Perth. So, it was really hard going"*—P4

### 3.1.3. The Cost Is Greater

Other factors that impacted the participants' cancer experience included the financial demands of travel, the cost of treatment and living expenses, which caused frustration and worry. Participants two and three describe the financial limitations with the cost of accommodation and meals adding to an already stretched household budget.

*"And you just couldn't afford the hotel prices down there. I mean, it's just expensive. It's enough having to spend extra on your food and drink and getting to your house from the supermarket trip or the taxis or the buses."*—P2

*"It is a bit more of a hassle. And obviously sometimes you can't get the cheaper hotel or sometimes you don't want the cheaper hotel because it's not very nice. And so you begin somewhere else and end up paying 30, 40 dollars a night out of pocket."*—P3

Cancer Council Western Australia accommodation was the first option for all participants due to its convenient location and ease of access to appointments. Added stress was noted among participants who had to seek alternative accommodation which was expensive and difficult to access, as described by participants one and nine.

> *"And the first couple of times we went to Perth we stayed at motels and I couldn't get into Crawford lodge. It's easy if you book in advance. It is so in demand, especially for country people."*—P1

> *"The hotel...it was hot, it was noisy, and going through treatment was just horrendous. Having to go up and down stairs...I could hardly breathe as it was, let alone climb stairs."*—P9

*3.2. Theme 2: The Paradoxical Experience of Staying at the Accommodation*

This theme explores participants' experiences while living in CCWA accommodation, the facilities and services provided by CCWA, as well as external supports, and how these enabled participants to continue engaging in meaningful activities. Paradoxically for some participants, the experience meant they found it difficult to adjust to their new living environment.

3.2.1. How the Accommodation Helped Me to Engage in Activities

Participants commended the quality and range of facilities within CCWA accommodation that enabled them to engage in meaningful activities including access to the garden and kitchen. Conversely, a few participants reported that the communal kitchen prevented them from cooking during times when they did not want to interact with others or were feeling unwell.

Participants were able to engage in leisure activities such as doing puzzles, having morning and afternoon teas in the lounge area, or watching a movie in the theatre. Others utilised the gymnasium and exercise classes. The availability of a business area, equipped with computer and printing facilities, enabled participants to continue engaging in work-related tasks even while away from their offices. Participants four and five describe the social benefits and the opportunity to participate in usual roles and routines

> *"I'm impressed with the fact that they try to make it as inclusive as possible for people who are there for longer times like with the morning and afternoon teas, the social get together. I actually did go to the exercise classes as well.*

> *"So, I have a bit of a [vegie] patch outside and water the garden. So, they put these in Milroy that mimicked my normal day."*—P5

3.2.2. Practical and Social Supports Offered by CCWA and External Services

Cancer Council Western Australia provided various services, which had positive effects on the participants' cancer experience. Having the staff check in on them and other 'guests' helped foster a "family spirit" at the accommodation. This created a feeling of ease and safety as participants felt accepted and supported, knowing that they were "all there for the same reason." Participant three particularly valued the frequent support offered by the support coordinator, who was available at the accommodation service.

> *"So, I think the nurse (cancer support coordinator) came in and checked on me once or twice and then call every couple of days just to make sure I was okay."*—P3

Participants spoke highly of their experiences with various external services. The volunteer-run "Look Good Feel Better" service aims to improve consumer's confidence with accessories, makeup, and outfits. The Solaris Centre, located in the main hospital, is a volunteer service that provides an array of complimentary therapies. The Patient Assistance Transport Scheme (PATS) is a government-funded program that provides eligible Australian residents residing in rural locations with financial assistance to access treatment away from

home as well as social support and symptom management. Participant three appreciated the assistance with arranging flights and accommodation.

> *"Can you just book the flights? Can you just book the lodge? Can you just take care of everything? And they did so, all of those technicalities of traveling 2000 kilometres were taken care of."*—P3

### 3.2.3. How Life within CCWA Accommodation Was Different from the Usual Routine at Home

Participants reported that the routine experienced while at the accommodation was very different. This created mixed feelings of being well supported but not being able to complete their usual activities and routines. Participants three and four noted boredom due to the change of location and responsibilities.

> *"Oh, everything was turned upside down and I couldn't work. I couldn't do anything. . . . But then for the rest of the day, I had nothing to do and just have to kind of find ways to fill my time. Whereas at home I would be busy going to groups and doing all sorts of different things and working and taking care of the kids."*—P3

> *"But I think it was probably easier to control there because I didn't have to do a lot of meal preparation that was basically just taking the meal out of the freezer and then taking it down to the kitchen and zapping it in the microwave. So, it wasn't the same as is at home."*—P4

### 3.3. Theme 3: Feeling Grateful for Support Offered

Participants had the opportunity to reflect on their cancer journey post-treatment and reported a change of perspective regarding their experiences. Three subthemes were explored: gratitude for services provided, the impact of having support from family and friends, and suggestions to improve service provision.

### 3.3.1. Showing Gratitude for the Accommodation Services Provided

Some participants highlighted the significant impact CCWA accommodation had on their cancer treatment in Perth. CCWA fostered a sense of "family" within its accommodation, which encouraged participants to give back, allowing others to have the same experience. Participants reported donating gardening supplies, hand-crafted items, and monetary donations. These donations enable CCWA to provide the same or even improved level of service to future guests, ensuring continuation of the service.

> *"I got interested in the little vegie patch there. So I did some donations of some compost and some other stuff to go there because somebody else had sort of instigated getting the garden and had been running it."*—P5

> *"I mean, you know, I know they might be struggling now but I gave them a donation anyways."*—P1

### 3.3.2. Social Support from Family and Friends Was Invaluable

Support from family, friends and others living in the participant's local community played a significant role in mitigating some of the barriers discussed previously including assistance with pets, vehicles, homes and family members allowing them to focus on treatment. Some participants relied on others with similar experiences to assist them emotionally. Participants whose family or friends were with them during treatment stated this support was invaluable.

> *"My sister looked after my dog...My mechanic looked after my vehicle, my four-wheel drive...My mum would ring me every day at the Lodge...I didn't really have anything to worry about."*—P7

> *"And like I said, my parents came out here as well to support and my in-laws came out to Broome to support us as well. So, during the month that I had to be gone, they*

*were there as well to help for that time and afterwards to help with the kids and with my husband."—P3*

## 4. Discussion

The findings of this research demonstrate that temporarily relocating to the city from a rural or remote location for treatment impacts the individual with cancer due to a broad range of emotional, practical, social and financial factors. The availability of the accommodation services mitigated the severity of these factors, highlighting the pivotal role the service played. The supports offered by the accommodation service emphasised the need and value of the service.

The literature reports many similar barriers for people with cancer and their families [22]. These barriers include the side effects of treatment, such as appetite loss and nausea, change in energy levels, and fatigue. However, as reflected in the current findings, individuals from rural and remote areas face additional obstacles in their cancer experience when temporarily relocating to the city for treatment. Distress associated with separation from their community, family, and home (e.g., loneliness and sadness) as well as extra out-of-pocket expenses from medical and non-medical needs were notably challenging for participants. Earlier studies have documented similar negative emotional, financial, and social implications of accessing cancer services away from their home [9,10,23–25]. The study conducted by Butow et al. in 2012 also reported that several participants perceived the temporary relocation as a welcomed break from their everyday demands and responsibilities back home [24]. Some participants in our study shared this perspective and regarded their stay in the city as a "holiday" and an opportunity to remove themselves from some of their usual stressors.

Social support was discussed as a major component of the experience among the participants. The availability of social support is pivotal in the wellbeing of individuals with cancer and is seen to be significant in providing much needed hope [26]. Results from a longitudinal study showed that physical wellbeing is positively correlated to the level of social support available [27]. Our study indicates that support from family and friends helped them to manage the process of moving to the city for treatment. These supports were available in practical, social and emotional forms. Participants trusted friends and family in their home locations to attend to their affairs, which allowed them to focus on their treatment. A "family spirit" was fostered between the participants, staff and other guests within CCWA accommodations which brought about positive experiences for the participants. Furthermore, the accommodation service provided an unparalleled access to organisations which assisted participants to access various supports.

The results revealed a range of different experiences for guests staying at CCWA accommodation. Many participants highly valued the facilities that supported them to attend to basic care needs, desired activities and promoted social interactions. Despite being far from home, participants adapted to their environment and found a sense of normality by engaging in meaningful occupations that replicated their normal routine. People with chronic illness are resilient and can make adaptions to their lifestyle and social supports, which is crucial for developing self-empowerment and social capital [28,29]. Prioritising engagement in meaningful occupations, routines and adapting everyday tasks are key components in obtaining a return to normalcy [30].

In contrast, while the accommodation service permitted the continuation of meaningful occupations, temporarily relocating for treatment brings inevitable disruption to the usual routine that hindered the participants' experience. Occupational roles are a set of culturally and socially agreed upon behaviours that comprise an individual's primary life habits, routines, and rituals [31]. For some participants, being away from home was an unwelcomed break from the everyday demands of ordinary life as they were unable to perform their roles as a parent, spouse, or worker. Existing literature have also highlighted that being away from home can increase feelings of isolation, loss or function, self-identity and daily routine [26,32].

There is limited evidence on the experiences of people receiving services such as those offered by CCWA. Our study has revealed new insight into the nature of the accommodation service provision, the environment created and the impression it left on the participants. Feeling comfortable and safe while undergoing treatment is likely to assist people with cancer feel at ease and in control [28,33]. All participants were grateful for the support they received from the service. Some participants expressed their gratitude by giving back to the accommodation via donations. These donations enabled the accommodation service to provide the same or improved level of service to future guests, ensuring the continuation of the service.

COVID-19 has caused immense strain in healthcare services and is an ongoing unprecedented concern globally [34]. The pandemic has substantially impacted the trajectory of treatment for people with cancer and generated additional stress for this highly vulnerable group due to being immuno-compromised [34]. Participants staying at CCWA accommodation during the peak of COVID-19 had to follow social distancing restrictions and were unable to engage in many activities including basic needs such as grocery shopping. People with cancer already face unique challenges, therefore during such a crisis it is imperative that patients and their caregivers are supported with a safe environment and supportive care to alleviate the psychosocial impact [34].

The emotional, practical and social supports provided by the accommodation service can assist to alleviate some of the concerns associated with temporarily relocating to the city for treatment. Thus, having access to suitable accommodation is essential to improve their quality of life during this critical time by helping guests engage in meaningful occupations and meet their daily needs in alternative living situations. Our study highlights the need for accommodation services to continue providing essential resources and offer additional social, occupational and emotional supports for people undergoing treatment for cancer.

Limitations of this study include generalisability of findings. The study was conducted in a single setting. Despite CCWA having two accommodation facilities, most participants were from the lodge located within the main hospital. Participants were predominantly females and between the ages of 55 and 65. Questions pertaining to the participants' racial and ethnic backgrounds were not asked and it is suggested that future studies could include this information. Potential exploration of these issues backgrounds could offer further comparison and contrast to the experiences of these individuals during their relocation for cancer treatment. These factors suggest that the range of experiences documented in our findings cannot be concluded as exhaustive. The study could be strengthened by exploring the experiences of guests from the other lodge, males, younger and older individuals, people from other rural and remote areas accessing a different metropolitan cancer accommodation, as well as people at different stages of cancer. Although our findings do not reflect the experiences of all people with cancer, previous research accentuated the additional burden people from rural and remote areas face in comparison to people from metropolitan areas [4,6,10,11]. This study took place during the COVID-19 pandemic. During this time, participants staying at CCWA accommodation experienced heightened anxiety levels as many were immune suppressed. Social distancing rules and uncertainties surrounding the virus prompted participants to reduce their interactions with others and isolate themselves. Shared spaces and services with the accommodation, such as the cinema were also closed. Due to COVID-19 restrictions, participants were interviewed via phone or video call instead of face-to-face. The change in data collection may have influenced participants' willingness to share. Despite these limitations, the study has multiple strengths. A pilot interview was conducted to streamline the interview guide and process. Prior to interviews, researchers attended cancer-specific distress training organised by CCWA to learn how to manage potential distress in participants.

## 5. Conclusions

Australia has an ageing population, and with the incidence of cancer increasing as people age, the demand for suitable care and supports, together with the associated

economic costs, cancer is a growing concern for the Australian community [35]. It is predicted that in 2025, there will be over 17,000 new cancer cases and 5500 deaths from cancer [36]. The proportion of people living in rural and remote areas of WA are more isolated when compared to other developed parts of the world [37]. People diagnosed with cancer who have to relocate during treatment require emotional, logistical and social supports. Cancer accommodation services are essential in enabling individuals to continue engaging in meaningful occupations and maintain their quality of life. The findings of this study indicate that further investigations of the unique experiences of people with cancer who live in rural and remote locations is warranted.

**Author Contributions:** All authors; A.C., L.L.P., J.S., E.N. and S.K. provided equal contribution for all aspects of the work. A.C.: Conceptualisation, methodology, data collection, analysis and writing draft manuscript, L.L.P.: Conceptualisation, methodology, data collection, analysis and writing draft manuscript, J.S.: Conceptualisation, methodology, data collection, analysis and writing draft manuscript, E.N.: Conceptualisation, methodology, data collection, analysis and writing draft manuscript, S.K.: Conceptualisation, methodology, data collection, analysis and writing draft manuscript. All authors have read and agreed to the published version of the manuscript.

**Funding:** This research received no external funding.

**Institutional Review Board Statement:** Ethical approval was granted from the Curtin Human Research Ethics Committee (HREC) in December 2019. With changes in the method of data collection due to COVID-19, an amendment was approved in February 2020 (HREC number: HRE 2020-0030).

**Informed Consent Statement:** Due to the restrictions brought about by COVID-19, written consent was difficult to attain, therefore consent was obtained verbally at the beginning of each interview as part of audio recording. Consent for publication (include appropriate statements): Verbal consent was obtained, as per previous information.

**Data Availability Statement:** The data presented in this study are available on request from the corresponding author. The data are not publicly available due to privacy reasons.

**Conflicts of Interest:** The authors declare no conflict of interest.

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
