# Peer review of "Experiences of People with Cancer from Rural and Remote Areas of Western Australia Using Supported Accommodation in Perth While Undergoing Treatment"

_curroncol, doi:10.3390/curroncol29020101_

Round 1

Reviewer 1 Report

The authors present a qualitative study of the experiences of 10 patients from remote/rural communities in Australia who were able to stay at well-supported accommodations provided specifically for people with cancer undergoing treatment in Perth. I do think it's important to highlight the value of this kind of practical/psychosocial intervention, particularly for people living in remote/rural communities and the themes that emerged are well-presented.

I have some suggestions:

  • It's not entirely clear from the introduction how novel this research is. The discussion lists a number of papers with related findings, but there isn't a lot of comparison and contrast between this paper and others in the literature. It would be interesting to know what themes are common and which are more unique to this population.
  • Both in the title and a few places in the manuscript, I found the phrase "people with cancer and their families from rural and remote areas" to be awkward. One can figure out what is being communicated, but the phrase alone could mean all people with cancer who have family members in remote areas. I suggest finding a way to reword the title and other places where this appears.
  • Qualitative research is not my expertise, but 10 participants feels like a relatively low number. The authors describe recruitment coming to an end because of saturation of information/the absence of new themes from new participants. However, they also note in the discussion that male patients, younger patients, and some other demographics are not very well represented. That makes me wonder if themes really were exhausted.
  • Participants were diagnosed with cancer within the last 3 months, but "People with a more recent diagnosis were excluded". Three months is relatively recent. Clarification is required.
  • I don't understand what is described by "Two interview guides, one for current and one for past participants". 
  • In 2.4 "analysis by described by" just needs a simple correction.
  • My most serious concern about this paper is that I'm not sure it achieves adequate confidentiality for participants. Table 1 lists geographic location of each participant and quotes in the text indicate which participant they came from. I'm assuming some of these places are relatively small (a Google search suggests there are 3000 people in Derby). I think it's conceivable that with the geographic location, the time period during which data were collected, age group, marital status, and employment status, a friend or family member of a participant could work out who a particular quote was from. I suggest being less specific about geographic location (maybe small/medium/large population size and <500km, 500-1000km, 1000-2000km, etc.)

Author Response

Response to reviewers’ feedback

Feedback Reviewer 1

Response

  • It's not entirely clear from the introduction how novel this research is. The discussion lists a number of papers with related findings, but there isn't a lot of comparison and contrast between this paper and others in the literature. It would be interesting to know what themes are common and which are more unique to this population.

Thankyou for your extensive feedback, please find our response and edits

Paragraphs 1 and 2 discuss the parallels between our findings and other studies (emotional, social, financial and physical).

Our study also discussed the value of having friends to assist with managing affairs at home. We discussed a similar finding by Butow (2012) in our study that describes how participants found relocation to be a beneficial ‘break’ from the usual routine.

A unique finding discussed in the Discussion is the sense of a ‘family spirit’ being created, which we have explored. Paragraph 4 explores a further unique finding; and relates to the value of engaging in meaningful occupations while undergoing cancer treatment and this is supported by the citations to occupational therapy literature. We explore the limited evidence relating to the experiences of relocating for treatment and suggest this is na area for further development in the research.

We hope this is sufficient information to address this question?

  • Both in the title and a few places in the manuscript, I found the phrase "people with cancer and their families from rural and remote areas" to be awkward. One can figure out what is being communicated, but the phrase alone could mean all people with cancer who have family members in remote areas. I suggest finding a way to reword the title and other places where this appears.

In the interest of using ‘person first’ language, this phrase has been used to describe a person with the diagnosis of cancer rather than a ‘cancer sufferer’. Would a change in the title to remove reference to families make this clearer?  Eg:

‘Experiences of people with cancer from rural and remote areas of Western Australia using supported accommodation in Perth while undergoing treatment’. All instances of the full phrase have been removed.

  • Qualitative research is not my expertise, but 10 participants feels like a relatively low number. The authors describe recruitment coming to an end because of saturation of information/the absence of new themes from new participants. However, they also note in the discussion that male patients, younger patients, and some other demographics are not very well represented. That makes me wonder if themes really were exhausted.

It is our understanding that in qualitative analysis, there are no specific minimum participants to determine that saturation has occurred. In this study, analysis of data occurred alongside recruitment and interviews of participants (concurrent). Page 2 states ‘Recruitment ceased when no new concepts were identified during data analysis and   saturation of the data was considered to have been reached’. Section 2.4 also describes this process. For the purpose of this study we are only able to describe the experiences of the participants recruited. We also state on page 9 that male younger patients were not recruited and that this is a limitation of the study. This is further stated as follows ‘These factors suggest that the range of experiences documented in our findings cannot be concluded as exhaustive. The study could be strengthened by exploring the experiences of guests from the other lodge, males, younger and older individuals, people from other rural and remote areas accessing a different metropolitan cancer accommodation, as well as people at different stages of cancer’.

  • Participants were diagnosed with cancer within the last 3 months, but "People with a more recent diagnosis were excluded". Three months is relatively recent. Clarification is required.

We attempted to explain in the manuscript that people who are recently diagnosed are vulnerable due to the demands of treatment, our participants also needed to relocate for treatment and deal with a multitude of concerns (including their own mortality) due to the diagnosis. For this reason, we felt that attempting to recruit people for our research study with a diagnosis date of less than 3 months could create an enormous extra burden on top of their existing vulnerabilities. This information has been added as follows on page 2

‘A diagnosis of cancer represents an extreme challenge to a person’s sense of mortality, with many barriers relating to the diagnosis. Therefore, people with a more recent diagnosis were excluded from this study to prevent causing them experiencing additional burden associated with participating in a research study and to reduce any unnecessary distress created by this.’

  • I don't understand what is described by "Two interview guides, one for current and one for past participants". 

Participants were recruited from two distinct time periods- those that had previously stayed in the accommodations (and had returned to their rural/remote locations) and those that were currently living at the accommodations. The question guides were slightly modified to ensure that exactly the same questions were being asked, but the language required was (slightly) different eg

‘When you stayed at the accommodation…’

‘While you are staying at the accommodation’

  • In 2.4 "analysis by described by" just needs a simple correction.

Corrected

  • My most serious concern about this paper is that I'm not sure it achieves adequate confidentiality for participants. Table 1 lists geographic location of each participant and quotes in the text indicate which participant they came from. I'm assuming some of these places are relatively small (a Google search suggests there are 3000 people in Derby). I think it's conceivable that with the geographic location, the time period during which data were collected, age group, marital status, and employment status, a friend or family member of a participant could work out who a particular quote was from. I suggest being less specific about geographic location (maybe small/medium/large population size and <500km, 500-1000km, 1000-2000km, etc.)

Thank you for this notation. We agree that the privacy of participants is essential. Table 1 has been modified to remove the names of Towns and the distance from the capital city is approximate only, therefore preserving the anonymity of the participants

Reviewer 2 Report

The manuscript details a qualitative interview study of a small sample of rural/remote WA residents travelling to receive treatment in Perth. It is well written and clear and provides good insight into the experiences of people living with cancer in isolated areas and the challenges they face with travel. Although interesting, the study does not provide a lot in terms of a novel contribution to the literature. This could be counteracted through a stronger justification for the purpose of the study both in terms of aims and implications. Some minor queries and suggestions for improvement are as follows:

Abstract: I notice there was not abstract in the copy of the paper I was provided.

Introduction:

First page, Line 17: This is also true for other states in Australia and around the world. WA is a great example of this, but I would suggest removing reference to Perth, WA here unless it is used as an example to highlight your point.

First page Line 24 to 27: Could the authors rephrase this sentence. It reads as if long travel is attributed to inadequate transport/low SES ect. whereas (I would argue) these are other factors that contribute/exacerbate poorer outcomes. 

The last paragraph on the first page could be more concise. Can I suggest the authors revise and take out some of the repetition about "limited options" and "having to relocate/travel for treatment"

The study aims could be a little more specific. What is the purpose of learning about the lived experience? Is it to identify needs, challenges, opportunities for improvement, future research?

Methods:

Page 2 line 57: It's great the the authors gave a clear brief description of the method for the naive reader. Can I just suggest revising this sentence to make it more specific to this study "this enabled researchers to explore in-depth how individuals or groups experience the phenomena under exploration". For example, taking out "or groups" and replacing "the phenomena under exploration" with the actual study topic.

Participants: Participants needed to have had a dx in the last 3 months, however, "People with a more recent diagnosis were excluded from this study"... can the authors please specify how recent was too recent?

Could the authors please provide a little more detail about the CCWA lodges. Are they free? Government subsidised? Available to anyone with cancer? Can support people stay? It would also be good to mention the patient travel subsidy scheme in WA in setting the scene for the study before reporting the findings regarding financial challenges.

I am surprised that data saturation was reached after just 10 participants - How many interviews were conducted that did not yield new information? Were analyses being conducted as data was being collected? The authors cite Aldiaba et al., 2018 - could they please provide a little more detail on how they followed the guidance in this article to make their decision?

Page 3, Line 96: "A thick description of methods was detailed" - I'm unclear on what this means. Could the authors please state more explicitly?

Results:

Table 1: The authors provide a detailed description of each participant. I wonder if it may be too detailed given that some participants live in very remote, low population areas. I don't suppose this might inadvertently identify someone? I'm sure the authors have considered this, but wanted to flag just in case.

Discussion:

I don't think that the impact on family members can be directly derived from this study - I would suggest removing this from the first sentence or clarifying how this interpretation was made.

Line 304 - 307: a new paragraph starts in the middle of a sentence

The discussion largely summarises results with little focus on implications/recommendations etc. Can I suggest the authors revise and 1) cut down on the amount of results they repeat and 2) provide more discussion on the implications of their findings. I think the latter will be easier to do if a the study purpose is more specifically stated in the introduction. For example, if the purpose of the study is to identify areas for improvement or an intervention need, then the discussion can focus more on making suggestions/recommendations around this.

The first paragraph of the conclusion reads very much like the introduction. Can I suggest revising or removing this.

Author Response

Feedback Reviewer 2

Abstract: I notice there was not abstract in the copy of the paper I was provided.

Now provided

Introduction:

First page, Line 17: This is also true for other states in Australia and around the world. WA is a great example of this, but I would suggest removing reference to Perth, WA here unless it is used as an example to highlight your point.

First page Line 24 to 27: Could the authors rephrase this sentence. It reads as if long travel is attributed to inadequate transport/low SES ect. whereas (I would argue) these are other factors that contribute/exacerbate poorer outcomes. 

The last paragraph on the first page could be more concise. Can I suggest the authors revise and take out some of the repetition about "limited options" and "having to relocate/travel for treatment"

The study aims could be a little more specific. What is the purpose of learning about the lived experience? Is it to identify needs, challenges, opportunities for improvement, future research?

This has now been modified to read as

‘These concerns are often exacerbated for those living in rural and remote areas and are required to relocate in order to receive treatment’.

This has now been modified to read as ‘Multiple factors contribute to these inequalities, including delayed diagnosis, geographical isolation and inadequate transport which may result in people with cancer needing to relocate in order to access cancer treatment’. 

Revised- with the following sentences removed

‘In particular, the impact of cancer and its treatment, as well as the lived experience of utilising accommodation services. Given the location of their home and probable effects on timely access to treatment services, the needs of people with cancer from rural and remote areas of WA are worth investigating’

The following sentence has been added ‘The findings from this research will assist to ensure unmet needs are addressed and services are developed to ensure these needs are met’.

Participants: Participants needed to have had a dx in the last 3 months, however, "People with a more recent diagnosis were excluded from this study"... can the authors please specify how recent was too recent?

Could the authors please provide a little more detail about the CCWA lodges. Are they free? Government subsidised? Available to anyone with cancer? Can support people stay? It would also be good to mention the patient travel subsidy scheme in WA in setting the scene for the study before reporting the findings regarding financial challenges.

I am surprised that data saturation was reached after just 10 participants - How many interviews were conducted that did not yield new information? Were analyses being conducted as data was being collected? The authors cite Aldiaba et al., 2018 - could they please provide a little more detail on how they followed the guidance in this article to make their decision?

The following information has been added to ensure that the reader understands why people who received a diagnosis of cancer less than 3 months previously were excluded

‘A diagnosis of cancer represents an extreme challenge to a person’s sense of mortality, with many barriers relating to the diagnosis. Therefore. people with a more recent diagnosis were excluded from this study to prevent them experiencing additional burden associated with participating in a research study and to reduce any unnecessary distress created by this’ 

The following sentences have been added to the paper on page 1 line 42-45.

‘Individuals with cancer who live more than 100 kms away from relevant services and are required to relocate to Perth for treatment are eligible to stay at one of the CCWA lodges. The lodges offer subsidised accommodation for both individual and family members. Individuals who are eligible for the Patient Assisted Travel Scheme (PATS) may receive further financial support to assist with accommodation costs and these are then offered free of charge in some instances.’

It is our understanding that in qualitative analysis, there are no specific minimum participants to determine that saturation has occurred. In this study, analysis of data occurred alongside recruitment and interviews of participants (concurrent). Page 2 states ‘Recruitment ceased when no new concepts were identified during data analysis and   saturation of the data was considered to have been reached’. Section 2.4 also describes this process, citing Braun and Clarke (2006) and the steps undertaken by the authors to complete this

Methods:

Page 2 line 57: It's great the authors gave a clear brief description of the method for the naive reader. Can I just suggest revising this sentence to make it more specific to this study "this enabled researchers to explore in-depth how individuals or groups experience the phenomena under exploration". For example, taking out "or groups" and replacing "the phenomena under exploration" with the actual study topic.

Agreed, we have effected these changes in the manuscript and it now reads

‘Semi-structured in-depth interviews were chosen as the data collection method as this enabled researchers to explore in-depth how individuals with cancer experience the additional burden of relocating for treatment’

Results:

Table 1: The authors provide a detailed description of each participant. I wonder if it may be too detailed given that some participants live in very remote, low population areas. I don't suppose this might inadvertently identify someone? I'm sure the authors have considered this, but wanted to flag just in case.

Agree that this could potentially be a privacy issue.

Table 1 has been modified to remove the names of Towns and the distance from the capital city is approximate only, therefore preserving the anonymity of the participants

Discussion:

I don't think that the impact on family members can be directly derived from this study - I would suggest removing this from the first sentence or clarifying how this interpretation was made.

Line 304 - 307: a new paragraph starts in the middle of a sentence

The discussion largely summarises results with little focus on implications/recommendations etc. Can I suggest the authors revise and 1) cut down on the amount of results they repeat and 2) provide more discussion on the implications of their findings. I think the latter will be easier to do if a the study purpose is more specifically stated in the introduction. For example, if the purpose of the study is to identify areas for improvement or an intervention need, then the discussion can focus more on making suggestions/recommendations around this.

The first paragraph of the conclusion reads very much like the introduction. Can I suggest revising or removing this.

Agreed, the title had been changed and references to family members experiences removed from the text. This sentence now reads ‘The findings of this research demonstrate that temporarily relocating to the city from a rural or remote location for treatment impacts the individual with cancer due to a broad range of emotional, practical, social and financial factors’.

Thank you for noting this error, we have corrected

Agreed-we have modified the discussion accordingly.

1)      Removed instances of repetition in the manuscript

2)      Added further implications of the fundings

Revised